# Waveform Design Method for Piezoelectric Print-Head Based on Iterative Learning and Equivalent Circuit Model

**DOI:** 10.3390/mi14040768

**Published:** 2023-03-30

**Authors:** Jianjun Wang, Chuqing Xiong, Jin Huang, Ju Peng, Jie Zhang, Pengbing Zhao

**Affiliations:** School of Mechano-Electronic Engineering, Xidian University, Xi’an 710071, China

**Keywords:** piezoelectric print-head, iterative learning, droplet deposition technology

## Abstract

Piezoelectric print-heads (PPHs) are used with a variety of fluid materials with specific functions. Thus, the volume flow rate of the fluid at the nozzle determines the formation process of droplets, which is used to design the drive waveform of the PPH, control the volume flow rate at the nozzle, and effectively improve droplet deposition quality. In this study, based on the iterative learning and the equivalent circuit model of the PPHs, we proposed a waveform design method to control the volume flow rate at the nozzle. Experimental results show that the proposed method can accurately control the volume flow of the fluid at the nozzle. To verify the practical application value of the proposed method, we designed two drive waveforms to suppress residual vibration and produce smaller droplets. The results are exceptional, indicating that the proposed method has good practical application value.

## 1. Introduction

Piezoelectric droplet deposition technology has been widely used in many fields, including additive manufacturing [1] and electronic devices [2], because of its low operating cost and high material deposition accuracy. However, new application fields such as printing electrochemical sensors and optical microelectromechanical systems devices require better droplet deposition quality. A typical single-nozzle PPH structure is shown in Figure 1, and it mainly consists of a piezoelectric tube and glass tubes. When a voltage is applied to the electrode, the piezoelectric tube is polarized and causes the glass tube to shift radially. The deformation in the glass tube structure can cause pressure vibrations within the fluid. The pressure drives the liquid to form droplets at the nozzle. A detailed description of the droplet formation process can be found in the literature [3].

The droplet-forming quality is determined by the physical properties of the liquid, the structure of the print-head, and the drive waveform [4]. For a specific fluid and PPH, it is difficult to adjust the structure of the PPH and the physical properties of the liquid. Thus, the appropriate design of the drive waveform is the most convenient way to improve the droplet-forming quality. Currently, several drive waveforms exist for PPH according to different application requirements. The most classic drive waveform is the STDW [5,6,7], which drives the droplet produced with the PPH to have the same radius as that of the nozzle. The jetting ability of the PPH could be improved by adjusting the pulse width of the STDW. However, after the droplet is pushed from the PPH, the residual pressure vibrations in the fluid cannot be quickly dissipated. If the next droplet is jetted before the residual pressure vibrations settle, the resulting droplet properties (radius and velocity) will be different from those of the previous droplet. This can degrade the droplet-forming quality. To suppress residual pressure vibrations, a negative trapezoid pulse, called the DTDW, is added after the STDW [8,9,10]. The effect of the first trapezoid pulse is to produce droplets, and that of the second trapezoid pulse is to suppress residual pressure vibrations. Although the DTDW can suppress the residual pressure vibrations to some extent, some residual pressure vibrations remain. Moreover, some additional complex waveforms can be used to produce small droplets without reducing the nozzle radius [11,12]. In general, these drive waveforms ultimately change the volume flow rate of the liquid at the nozzle of the PPH. Therefore, controlling the volume flow rate of the fluid at the nozzle can improve the droplet-forming quality.

It is optimal to use an iterative learning method to improve the droplet-forming quality of PPHs because the droplet-forming process is executed repeatedly during material deposition. Wassink first applied the control method to PPHs to improve their jet frequencies [13]. Given the lack of a theoretical model for PPHs at the time, he identified a frequency response function as the theoretical model of the PPH based on the self-sensing signal [14]. However, since the self-sensing signal is usually composed of structural vibrations and pressure waves, the accuracy of the identified theoretical model was relatively low. In this study, based on the equivalent circuit model proposed in [15], we devise a new iterative, more practical learning waveform design method. Moreover, the existing iterative learning law for PPHs cannot be applied directly. This is because PPHs have two requirements for the drive waveform. The first requirement is that the drive waveform must have a limited driving cycle, and the second requirement mandates that the starting and ending voltages of the drive waveform must be zero. To satisfy these two requirements, some auxiliary constraints are added to the iterative learning process without changing the convergence. To verify the effect of this method, for the MJ-AL-80 PPH, we designed two types of injection volume flow rates of the fluid at the nozzle to suppress residual vibrations and generate smaller droplets. The optimized waveforms corresponding to the two flow rates are obtained using iterative learning. Finally, residual vibration suppression and smaller droplet-generating experiments were conducted based on the DWS [16]. The experimental results indicate that the proposed method can accurately control the volume flow rate at the nozzle.

The remainder of this paper is organized as follows. In Section 2, the equivalent circuit model and parameter estimation method are presented. In Section 3, we describe the iterative learning process. In Section 4, a schematic of the experimental system is introduced. Moreover, to suppress residual pressure vibrations and produce smaller droplets, two drive waveforms were designed based on the method proposed in this study, and the effectiveness of the method was verified using DWS. Finally, Section 5 summarizes this paper.

## 2. Model and Desired Volume Flow Rate

### 2.1. Equivalent Circuit Model

The structure of a circle squeeze-mode print-head MJ-AL-80 (a type of PPH fabricated by MicroFab; “MJ-AL” is the name of a series of PPHs, wherein “80” indicates the diameter of the nozzle in µm) is illustrated in Figure 1. The working principle of this PPH is that the drive waveform polarizes the piezoelectric tube to change the diameter of the pressure chamber. The volume change in the pressure chamber leads to a change in the inner fluid density, which causes pressure vibrations in the glass tube. This pressure vibration propagates to the nozzle and drives the fluid to produce droplets. In this process, the diameter change in the glass tube is very small (nanometer scale). To simplify the modeling process, the change in liquid density caused by the diameter change in the pressure chamber can be equivalent to the change in density caused by the increase or decrease in the inner fluid without a change in diameter.

The fluid injection process inside the PPH contains complex flow characteristics. Although the CFD theory can accurately simulate this process, the calculation process is very time-consuming, and the theoretical model in the differential form is difficult to combine with the control theory. The equivalent circuit model corresponds the fluid pressure and volume flow rate to the voltage and current in the circuit. Using a more mature circuit theory, the fluid injection dynamics can be expressed as a state equation. Not only is the solution efficiency very high, but the state equation can be fleetly combined with the control theory; therefore, the equivalent circuit model of the PPH is prepared in this study. Figure 2 illustrates the equivalent circuit of the MJ-AL-80 print-head. Capacitance *C*_1_ represents the compression in the fluid volume at the pressure chamber. The compressibility of the liquid in the right channel is equivalent to capacitance *C*_2_. The compressibility of the liquid in the left channel is ignored because the flow resistance at the exit is very small. The flow resistance and inertial force of the left channel and left half of the pressure chamber are equivalent to resistance *R*_1_ and inductance *L*_1_, respectively. Analogously, the flow resistance and inertial force of the right channel and right half of the pressure chamber are equivalent to resistance *R*_2_ and inductance *L*_2_, respectively. The flow resistance and inertial force of the nozzle are equivalent to resistance *R*_3_ and inductance *L*_3_, respectively. The ambient pressure is equivalent to voltage source *Us* (typically *Us* = 1 [atm]) [17]. The voltage source *Ud* is used to replace a pressure constant derived from the equation of state. Capacitance *C*_3_ represents the fluid stored by the meniscus surface tension [18]. The current source *i_s_* is used to simulate the increase or decrease in the fluid inside the pressure chamber.

According to KVL and KCL, the equivalent circuit illustrated in Figure 2 can be described using the following state equation:(1)$$\left\{ \begin{array}{l} \dot{x}=Ax+Bu\\ y=Cx \end{array}\right.$$



$$A=\left[\begin{array}{cccccc} 0 & 0 & 0 & \frac{1}{C_{1}} & -\frac{1}{C_{1}} & 0\\ 0 & 0 & 0 & 0 & \frac{1}{C_{2}} & -\frac{1}{C_{2}}\\ 0 & 0 & 0 & 0 & 0 & \frac{1}{C_{3}}\\ -\frac{1}{L_{1}} & 0 & 0 & -\frac{R_{1}}{L_{1}} & 0 & 0\\ \frac{1}{L_{2}} & -\frac{1}{L_{2}} & 0 & 0 & -\frac{R_{2}}{L_{2}} & 0\\ 0 & \frac{1}{L_{3}} & -\frac{1}{L_{3}} & 0 & 0 & -\frac{R_{3}}{L_{3}} \end{array}\right],B=\left[\begin{array}{cc} \frac{1}{C_{1}} & 0\\ 0 & 0\\ 0 & 0\\ 0 & \frac{1}{L_{1}}\\ 0 & 0\\ 0 & -\frac{1}{L_{3}} \end{array}\right],\;C={\left[\begin{array}{cccccc} 0 & 0 & 0 & 0 & 0 & 1 \end{array}\right]}^{T},u={\left[\begin{array}{cc} Us+Ud & i_{s} \end{array}\right]}^{T},x={\left[\begin{array}{cccccc} u_{1} & u_{2} & u_{3} & i_{1} & i_{2} & i_{3} \end{array}\right]}^{T}$$



In Equation (1), **A** is the state matrix, x is the state variable, **B** is the input matrix, **u** is the system input, **C** is the output matrix, and **y** is the system output.

### 2.2. Parameter Estimation

According to electroacoustic theory, the resistance, inductance, capacitance, and voltage source in the equivalent circuit shown in Figure 2 represent the viscous force, inertial force, compressibility, and pressure in the fluid, respectively [17]. The parameter estimations of resistance and inductance can be undertaken using Equations (2) and (3), respectively,
(2)$$R_{f}=\sqrt{\frac{2c\rho \mu l^{2}}{l_{T}\pi r^{6}}}$$
(3)$$L_{f}=\frac{l\rho }{\pi r^{2}}$$
where *c* is the speed of sound in the fluid, *ρ* is the density of the fluid, *µ* is the viscosity of the fluid, *l* is the length of the estimated portion of the channel, *l_T_* is the total length of the channel, *r* is the inner radius of the channel, and *R_f_* and *L_f_* are the equivalent resistance and inductance of the estimated portion of the channel, respectively.

As the droplet deposition process is very transient, it can be assumed to be an adiabatic process [19], and the pressure change in the pressure chamber can be described by the state equation (*dp* = *c*^2^*dρ*) under adiabatic conditions. The state equation of the pressure chamber can be described as
(4)$$dp=c^{2}\rho (t_{0})-\frac{c^{2}\rho (t_{0})\left[V(t_{0})+\int_{t_{0}}^{t}q(t)dt\right]}{V(t)}$$
where *dp* is the increase in pressure, *ρ*(*t*_0_) is the initial density of the liquid in the pressure chamber, *q*(*t*) is the volume flow rate of the fluid in the pressure chamber, and *V*(*t*) is the volume of the pressure chamber. As the volume change in the pressure chamber is very small, the change in liquid density caused by the diameter change in the pressure chamber can be simulated using the fluid source *q_s_*(*t*). Then, Equation (4) can be written as follows:(5)$$dp=c^{2}\rho (t_{0})-\frac{c^{2}\rho (t_{0})\left(V(t_{0})+\int_{t_{0}}^{t}q(t)dt+\int_{t_{0}}^{t}q_{s}(t)dt\right)}{V(t_{0})}$$
where *V*(*t*_0_) is the initial volume of the pressure chamber. To establish the relationship between *V*(*t*) and *q_s_*(*t*), let Equations (4) and (5) have the same pressure increment. Therefore, the relationship between *V*(*t*) and *q_s_*(*t*) can be described as follows:(6)$$q_{s}(t)=\frac{\partial V(t)}{\partial t}$$

According to Equation (5), the pressure in the pressure chamber can be expressed as the sum of the initial pressure and the pressure increment:(7)$$p(t)=p(t_{0})+c^{2}\rho (t_{0})-\frac{V(t_{0})+\int_{t_{0}}^{t}q(t)dt+\int_{t_{0}}^{t}q_{s}(t)dt}{\frac{V(t_{0})}{c^{2}\rho (t_{0})}}$$
where *p*(*t*_0_) is the initial pressure in the pressure chamber. In the equivalent circuit illustrated in Figure 2, voltage *u*_1_ represents the pressure change in the pressure chamber. According to the Kirchhoff Voltage Law (KVL), *u*_1_ can be written as
(8)$$u_{1}(t)=Us-Ud+\frac{Q_{1}(t_{0})+\int i_{c1}dt+\int i_{s}dt}{C_{1}}$$
where *i_c_*_1_ is the current through capacitance *C*_1_, *i_s_* is the current from the current source, and *Q*_1_(*t*_0_) is the initial charge on capacitor *C*_1_. This current source releases the charge to capacitance *C*_1_, causing voltage *u_c_*_1_ to increase. The current source absorbs the charge from capacitance *C*_1_, causing voltage *u_c_*_1_ to decrease. This process is very similar to that causing the pressure changes in the pressure chamber. Therefore, the parameter estimation method for capacitance and the voltage source *Ud* in the equivalent circuit can be respectively written as
(9)$$C_{f}=\frac{V(t_{0})}{c^{2}\rho (t_{0})}$$
(10)$$Ud=c^{2}\rho (t_{0})$$
where *C_f_* is the equivalent capacitance of the liquid in the pressure chamber. A capacitor *C_n_* is equivalent to the Laplace pressure caused by surface tension. To estimate the expression of the capacitor parameter, the average values of the flow-out/-in volumes at the nozzle are roof-shaped [17]. The equivalent capacitance of the surface tension can be described as
(11)$$C_{n}=\frac{V_{men}}{p_{men}}=\frac{\pi r_{n}^{4}}{3\sigma }$$
where *V_men_* is the roof-shaped volume of the fluid with nozzle radius *r_n_*, *p_men_* is the Laplace pressure, *σ* is the coefficient of surface tension, and *C_n_* is the equivalent capacitance of the meniscus. Based on the above equation, the estimation method of component parameters in the equivalent circuit illustrated in Figure 2 is listed in Table 1. A more detailed method of parameter estimation can be found in Reference [15].

## 3. Iterative Learning Process

### 3.1. Iterative Learning Method

As the PPH system is a causal system, the input of the drive waveform at any given time will always affect the output thereafter. It is difficult to obtain a good drive waveform using the classical proportional integral derivative (PID) iterative learning law. However, as the time scale of the drive waveform is limited to tens of microseconds, we can consider the system input vector as the object of iteration. This framework is called the lifted-system framework [20]. The discrete form of state Equation (1) can be written as
(12)$$y(k)=Gu(k)+d_{0}$$
where $d_{0}={\left[CAx(0),CA^{2}x(0),\cdots ,CA^{N}x(0)\right]}^{T}$ is the initial conditional response of the system, $y(k)={\left[y(1,k),y(2,k),\cdots ,y(N,k)\right]}^{T}$ is the system output vector corresponding to iteration *k*, $u(k)={\left[u(1,k),u(2,k),\cdots ,u(N,k)\right]}^{T}$ is the system input vector corresponding to iteration *k*, and $G=\left[\begin{array}{cccc} CB & 0 & \cdots & 0\\ CAB & CB & \cdots & 0\\ \vdots & \vdots & \ddots & \vdots \\ CA^{N-1}B & CA^{N-2}B & \cdots & CB \end{array}\right]$ is a Toeplitz matrix. **A**, **B**, and **C** are the discrete system state-transition matrix, input matrix, and output matrix, respectively. For a print-head system, the goal of the iterative learning waveform design method is to find system input **u** within a given time interval [0, N] so that the system output is consistent with the reference output (desired volume flow rate). According to Equation (12), the error vector **e**(*k*) corresponding to iteration *k* between the system output and reference output can be written as
(13)$$e(k)=y_{d}-Gu(k)-d_{0}$$
where $y_{d}={\left[y_{d}(1),y_{d}(2),\cdots ,y_{d}(N)\right]}^{T}$ is the system reference output vector. The quadratic form of the error vector corresponding to iteration *k* can be written as
(14)$$E(k)=\frac{1}{2}e{\left(k\right)}^{T}e(k)$$
where *E*(*k*) is the total error corresponding to iteration *k*. The purpose of the iterative learning waveform design is to find a system input **u** such that $\underset{k\to \infty }{\lim}E(k)=0$. To rapidly reduce the total error *E*, the steepest descent method is used to design the iterative learning laws [21,22]. The normalized negative gradient vector of the total error can be written as
(15)$$\left\{ \begin{array}{l} v=-{\left[\frac{\partial E(k)}{\partial u(0,k)},\frac{\partial E(k)}{\partial u(1,k)},\cdots ,\frac{\partial E(k)}{\partial u(N-1,k)}\right]}^{T}\\ \Delta u(k)=\frac{v}{{\left\|v\right\|}_{2}} \end{array}\right.$$
where Δ**u** is the normalized negative gradient vector of the total error and **v** is the negative gradient vector without normalization. Therefore, the learning law of the iterative learning system can be written as
(16)u(k+1)=u(k)+ηΔu(k)
where *η* is the iterative step length.

### 3.2. Optimal Iteration Step Length

In Equation (16), when *η* is too large, the learning process will not converge, and when *η* is too small, the convergence rate will be reduced. Combining Equations (13) and (16), the total error at time *k* + 1 can be written as
(17)$$E(k+1)=E(k)+\frac{1}{2}\left[\Delta u{\left(k\right)}^{T}G^{T}G\Delta u(k)\eta^{2}-2e{\left(k\right)}^{T}G\Delta u(k)\eta \right]$$

The second term on the right-hand side of Equation (17) is denoted as the total error increment Δ*E*(*k*) at iteration *k*. The expression for the total error increment Δ*E*(*k*) can be written as
(18)$$\Delta E(k)=\frac{1}{2}\Delta u{\left(k\right)}^{T}G^{T}G\Delta u(k)\eta^{2}-e{\left(k\right)}^{T}G\Delta u(k)\eta$$

According to Equation (18), the total error increment is a quadratic function with the iterative step length *η* as the independent variable. An iterative step length *η* must exist in each iteration cycle to minimize the total error increment Δ*E*. As *E*(*k* + 1) = *E*(*k*) + Δ*E*(*k*), the total error will decrease as fast as possible when the total error increment Δ*E* is equal to the minimum value in all the iteration steps. According to the extremum formula of the quadratic function, the optimal iterative step length *η* can be expressed as [21]
(19)$$\eta =\frac{e{\left(k\right)}^{T}G\Delta u(k)}{\Delta u{\left(k\right)}^{T}G^{T}G\Delta u(k)}$$

### 3.3. Adding to Iterative Learning Constraint

The aforementioned iterative learning process can be used to obtain a good drive waveform. However, the drive waveform of the print-head must meet some requirements in practical applications. The first requirement is that the drive waveform should be loaded within a finite amount of time. The second requirement is that the waveform starting and ending voltages must be zero. To satisfy the first requirement, a weight function, as shown in Figure 3, is used to ensure that the system input is valid for a limited period of time. The iterative learning law described in Equation (16) can be rewritten as
(20)$$u(k+1)=u(k)+\eta \Delta u(k)w^{T}$$
where **W** is also a vector defined in n-dimensional real space. This weight function makes iterative learning effective only for a limited amount of time.

The system input in Equation (1) consists of two parts: the first part is a pressure constant, and the second part is the volume flow rate of the fluid in the pressure chamber. The task of iterative learning is to find a change rule of the volume flow rate of the fluid in the pressure chamber to control the volume flow rate of the fluid at the nozzle. The volume flow rate of the fluid in the pressure chamber is equal to the derivative of the drive waveform multiplied by a coefficient. Therefore, the drive waveform can be obtained by integrating the volume flow rate obtained using iterative learning. To satisfy the second requirement, the second term on the right-hand-side of Equation (20) is modified using the following equation:(21)$$\left\{ \begin{array}{l} \begin{array}{cc} {\left(m\right)}_{P}={\left(m\right)}_{P}\left(\frac{\int {\left(m\right)}_{N}dt}{\int {\left(m\right)}_{P}dt}\right) & \int {\left(m\right)}_{P}dt>\int {\left(m\right)}_{N}dt \end{array}\\ \begin{array}{cc} {\left(m\right)}_{N}={\left(m\right)}_{N}\left(\frac{\int {\left(m\right)}_{P}dt}{\int {\left(m\right)}_{N}dt}\right) & \int {\left(m\right)}_{P}dt<\int {\left(m\right)}_{N}dt \end{array} \end{array}\right.$$
where **m** = *η*Δ**u**(*k*)**w***^T^* is the weighted iterative learning system input correction, and (**m**)_P_ and (**m**)_N_ represent the elements of the **m** vector that are greater than zero and less than zero, respectively. The purpose of Equation (21) is to set the integral of the system input correction *η*Δ**u**(*k*)**w***^T^* calculated in each iteration step to zero, which implies that the starting and ending voltages of the drive waveform obtained by integrating the volume flow rate are zero.

### 3.4. Calculated Waveform with the Iterative Learning Method

Based on the algorithm principle mentioned above, the control input corresponding to the desired injection volume flow rate can be iteratively computed according to the calculation process shown in Figure 4. Firstly, the expected injection volume flow rate at the nozzle is designed and set as the iterative optimization target **y***_d_*. Secondly, the model output **y**(*k*) corresponding to any initial system input **u**(0) is calculated, and the correction amount of the control input is calculated according to Equations (14)–(21). This process is repeated until the total error *E*(*k*) converges to the set threshold. Finally, the corresponding control input is converted into a voltage waveform, which generates the expected injection behavior when driving the PPH and thus optimizes the fluid injection performance.

## 4. Results and Discussion

### 4.1. Experimental Setup

We verify the actual effect of droplet formation by optimizing waveform generation. A schematic of the DWS is illustrated in Figure 5. This system mainly consists of a USB camera (MV-VD040SM, Microvision, Redmond, WA, USA), light-emitting diode (LED) strobe, PPH (MJ-AL-80, MicroFab, Plano, TX, USA), droplet watch system controller, pressure controller, and data processing software that runs on a personal computer. A USB camera with an external trigger interface is used to record the droplet images at different times. The LED strobe is used to control the exposure time of the USB camera because the default exposure time of the camera is so long that the edge of the captured droplet image is blurred. To reduce the effective exposure time of the USB camera, the lighting time of the LED strobe is controlled with the droplet watch system controller. When the LED strobe is dark, the USB camera cannot be effectively exposed even during the exposure time, because the incident light is insufficient. The exposure time of the USB camera can be adjusted by setting the lighting time of the LED strobe. To ensure that the edge of the droplet image captured with the USB camera is as clear as possible without reducing the contrast, the lighting time of the LED strobe should be as short as possible. The pressure controller generates negative pressure inside the PPH to balance the weight of the fluid. The data processing software is used to analyze the photos captured with the USB camera and extract motion information of the meniscus. The detailed experimental theory can be found in Reference [16].

### 4.2. Suppressed Residual Vibration

Residual vibration during droplet deposition is the primary factor limiting the deposition efficiency and quality. For ethanol, the volume flow rate corresponding to STDW is shown in the magenta triangle line (the line is obtained by solving the state equation) in Figure 6a, and the results show that there are multiple peaks. 

Except for the first peak, other peaks are redundant. To ensure that the new droplet ejection behavior is not affected by residual oscillation, it is necessary to wait for the residual peak energy to decay to zero before the new droplet ejection, which limits the ejection frequency of microdroplets. Therefore, removing residual peaks is crucial to maximizing injection frequency. In this study, we use a weighted window function (the blue line in Figure 6a) to remove residual peaks and obtain the desired injection volume flow rate (the red circle line in Figure 6a).

The desired injection volume flow rate in Figure 6a is used as the optimization target for iterative learning, and the iterative algorithms are started. Figure 6b shows the injection volume flow rate (solid blue lines) of each iteration step in the iterative learning process, the results show that as the iterations increase, the injection volume flow rate keeps approaching the optimization target, and at the 50th iteration step, the residual peak has been well suppressed (as shown by the magenta circle line in Figure 6b). Figure 6c shows the change in total error, *E*(*k*), and iteration step length, *η*(*k*), with the iterations. The results show that the total error in the iteration process is strictly convergent (it can be seen from the figure that the total error, *E*(*k*), basically converges after 10 iterations, which shows that the iterative algorithm has fast convergence characteristics and can improve the real-time performance of the controller), and the optimal iteration step length is automatically calculated according to the error. Figure 6d shows the optimized drive waveform. Compared with the STDW, the optimized drive waveform has a new negative voltage pulse, which is mainly used to suppress the residual pressure vibrations.

The experimental system described in Section 4.1 is used to verify the actual effects of suppressing residual vibrations. Figure 7a shows the meniscus vibration in the ethanol when the STDW is used. The results show that the vibration period of the meniscus is about 46 μs, and the volume is the largest at around 60, 106, 153, and 200 μs, respectively. This is highly consistent with the prediction results in Figure 6a, which proves that the model established in this study has high accuracy. Figure 7b shows the vibration in the meniscus when the optimized drive waveform is used. The results indicate that the optimized drive waveform presented can effectively suppress the residual vibrations. Moreover, when the STDW is used for MJ-AL-80 PPH, and the frequency is higher than 1 kHz, an unstable ejection phenomenon will occur (as shown in Figure 7c). When the optimized drive waveform is used, the droplet ejection process is still stable at the frequency of 7 kHz (as shown in Figure 7d). This shows that the optimized waveform can significantly improve the droplet ejection frequency. 

### 4.3. Production of Smaller Droplets

Typically, a droplet produced with the PPH has the same diameter as that of the nozzle. However, some special drive waveforms can be used to reduce the droplet diameter without reducing the nozzle diameter. These drive waveforms produce smaller drops by forming a tongue on the meniscus [11]. However, it is difficult to produce smaller droplets when the viscosity of the fluid is too low [12]. This is because it is difficult for low-viscosity fluids to form a stable tongue on the meniscus. However, as shown in Figure 8a, increasing the drive force of a low-viscosity fluid can produce multiple smaller droplets, and the second smaller droplet is clearly formed by trailing tails. One way to eliminate the second/following droplet is to create a new meniscus that pulls the tail back inside the nozzle. To create a new meniscus at the nozzle after the main droplet is formed, the volume flow rate with two volume flow rate peaks shown in Figure 8b is used. The first peak is used to produce smaller droplets, and the second peak is used to form a new meniscus at the nozzle to pull the tail back inside the PPH. Similar to the reference volume flow rate to suppress residual pressure vibration, to ensure the accuracy of calculation, the discretization period of this process is 1 μs. The volume flow rate in Figure 8b is used for iterative learning. Similar to Figure 6, the optimized waveform shown in Figure 8c can be obtained using iterative learning. Figure 8d shows the formation of smaller droplets when the optimized waveform in Figure 8c is used. When the main droplet breaks, a new meniscus is formed to pull the tail back inside the nozzle. The diameter of the PPH nozzle is 80 µm and that of the smaller droplet is approximately 26 µm. The droplet diameter decreases by 67.5%.

Taken together, the experimental results indicate that the proposed method can accurately control the volume flow rate of fluid at the nozzle. It can be used to suppress residual vibrations and produce smaller droplets. Moreover, this method can be applied to various other applications such as controlling the speed and size of the droplets.

## 5. Conclusions

This paper proposes a more practical waveform design method based on the equivalent circuit model and iterative learning, which can quickly calculate the drive waveform corresponding to the desired volume flow rate. To verify the practical effect of the proposed method, we designed the drive waveforms for suppressing residual pressure vibration and generating smaller droplets, respectively. For the suppression of residual pressure vibration, the experimental results show that the drive waveform designed with the proposed method can well remove the redundant pressure peak, and the injection frequency can be increased by 7 times. For the generation of smaller droplets, the experimental results show that the drive waveform designed with the proposed method can reduce the droplet volume of low-viscosity fluid by 67%. The experimental results indicate that this method can accurately control the volume flow rate of the fluid at the nozzle. However, the surface behavior of the fluid at the nozzle is complex, and the surface behaviors will change with different volume flow rates. For more complex applications, it is difficult to find a corresponding reference volume flow rate. Future studies will focus on this aspect.

## Figures and Tables

**Figure 1 micromachines-14-00768-f001:**
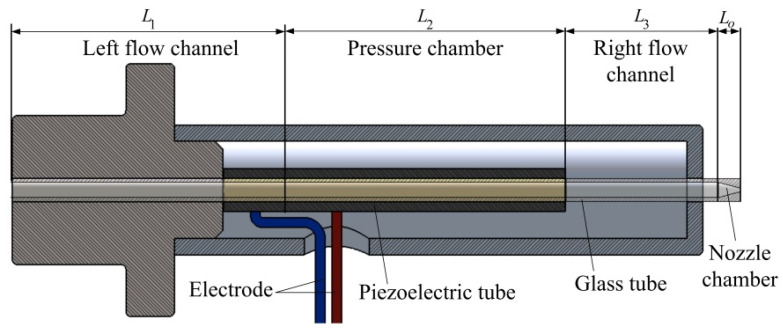
Structure of the MJ-AL-80 PPH.

**Figure 2 micromachines-14-00768-f002:**
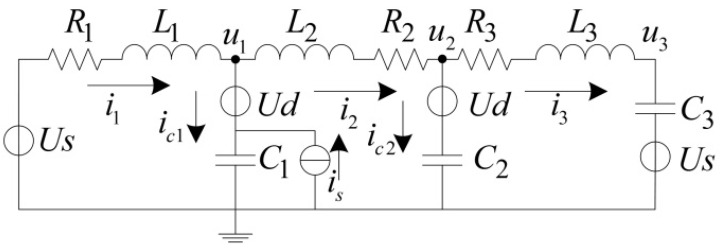
Equivalent circuit model of the inkjet print-head.

**Figure 3 micromachines-14-00768-f003:**
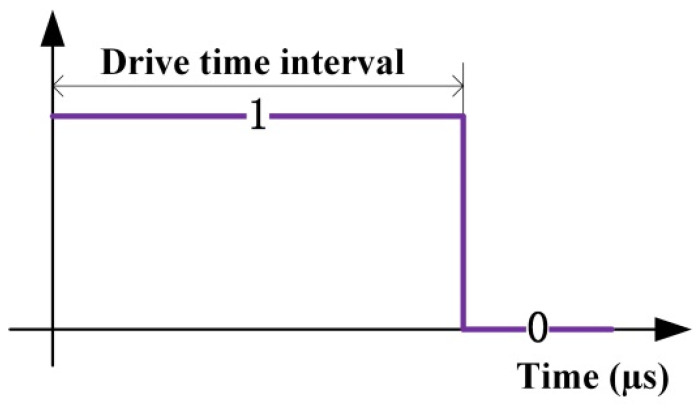
Weight function controlling the effective interval of iterative learning.

**Figure 4 micromachines-14-00768-f004:**
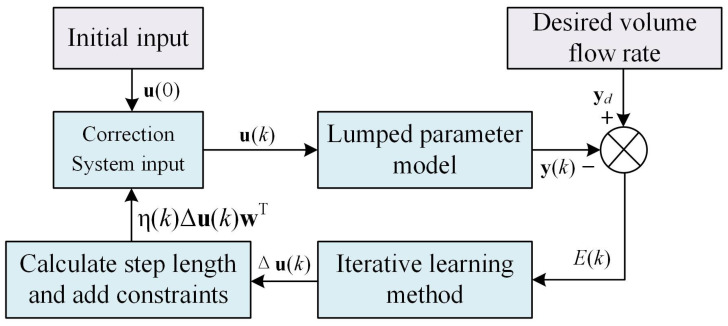
Principles of the iterative learning method.

**Figure 5 micromachines-14-00768-f005:**
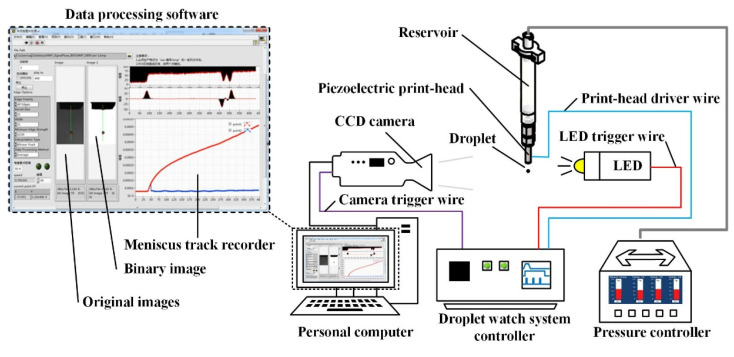
Schematic of the experimental system.

**Figure 6 micromachines-14-00768-f006:**
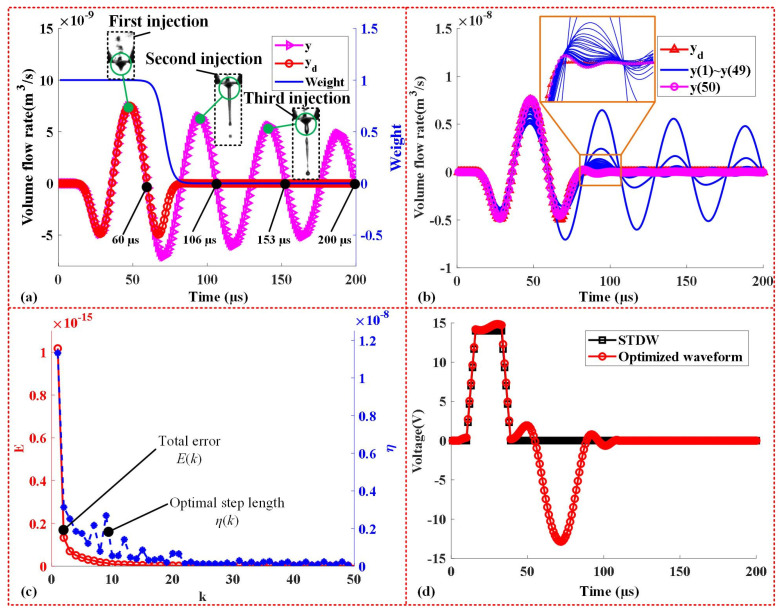
(**a**). Desired injection volume flow rate designed to suppress residual vibration. (**b**). The iterative learning process of the injection volume flow rate that inhibits the residual vibrations (**c**). The change in total error and optimal step length with iterations. (**d**). Optimal waveform for suppressing residual pressure vibrations.

**Figure 7 micromachines-14-00768-f007:**
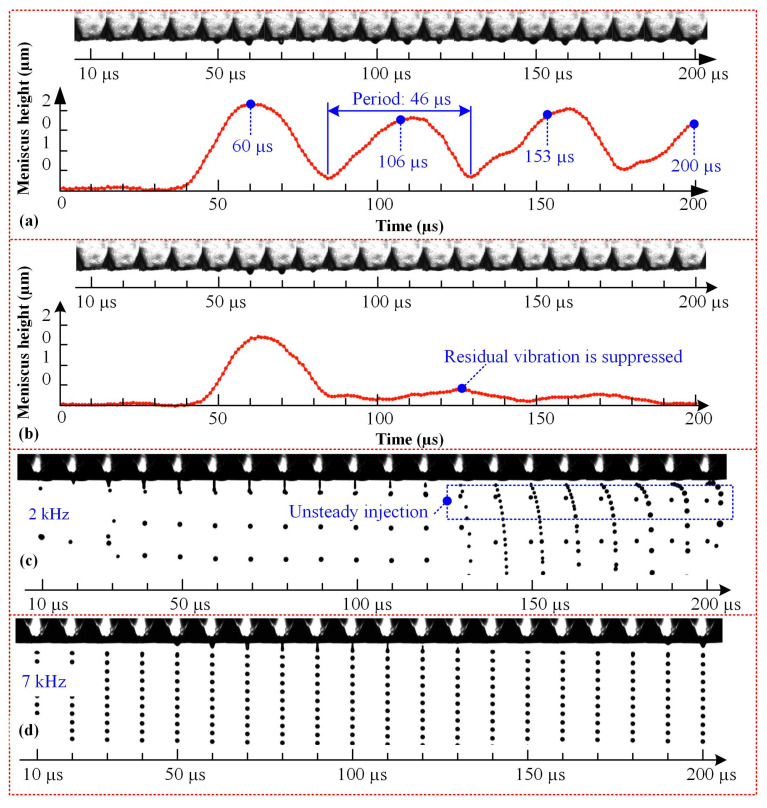
(**a**). Meniscus vibration process corresponding to STDW. (**b**). Meniscus vibration process corresponding to the optimized waveform. (**c**). At 2 kHz frequency, the droplet ejection process corresponding to STDW is unstable. (**d**). At 7 kHz frequency, the droplet ejection process corresponding to the optimized waveform is stable.

**Figure 8 micromachines-14-00768-f008:**
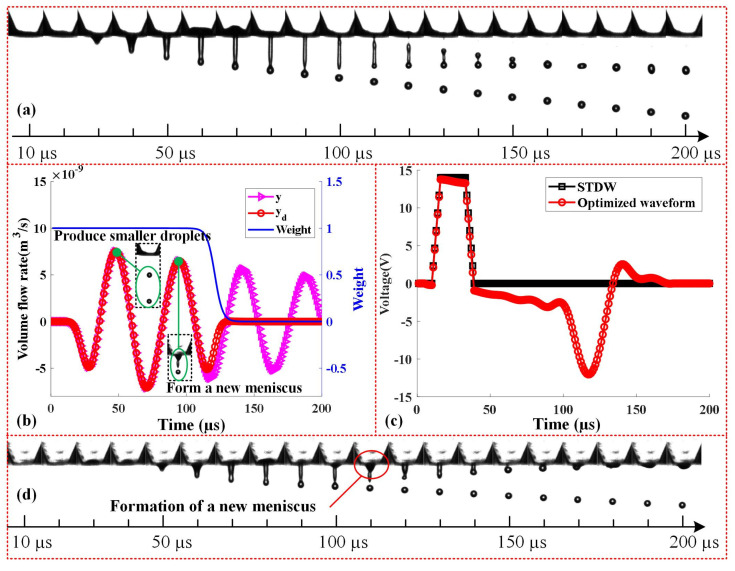
(**a**). Process of forming two smaller droplets (ethanol is used as the fluid). (**b**). Desired volume flow rate to produce smaller droplet. (**c**). Optimized waveform obtained using iterative learning. (**d**). Process of generating a smaller droplet using the optimized waveform.

**Table 1 micromachines-14-00768-t001:** Parameters mapping relation.

Equivalent Circuit Components	Mapping Expressions	Parameters for MJ-AL-80
Capacitance *C*_1_, *C*_2_, *C*_3_	$\frac{\pi r^{2}L_{2}}{c^{2}\rho (t_{0})}$ $\frac{\pi r^{2}L_{3}}{c^{2}\rho (t_{0})}$ $,\;\frac{\pi r_{o}^{4}}{3\sigma }$	*C*_1_ = 1.4374 × 10^−18^ *C*_2_ = 8.2565 × 10^−19^ *C*_3_ = 1.1888 × 10^−16^
Voltage source *Ud*	$c^{2}\rho (t_{0})$	*Ud* = 9.897216 × 10^8^
Resistance *R*_1_, *R*_2_, *R*_3_	$\sqrt{\frac{2c\rho \mu {\left(L_{1}+L_{2}/2\right)}^{2}}{l_{T}\pi r^{6}}}$ $,\sqrt{\frac{2c\rho \mu {\left(L_{3}+L_{2}/2\right)}^{2}}{l_{T}\pi r^{6}}}$ $,\sqrt{\frac{2c\rho \mu L_{o}{}^{2}}{l_{T}\pi r_{e}^{6}}}$	*R*_1_ = 2.3018 × 10^11^ *R*_2_ = 1.5635 × 10^11^ *R*_3_ = 3.7675 × 10^11^
Inductance *L*_1_, *L*_2_, *L*_3_	$\frac{\rho \left(L_{1}+L_{2}/2\right)}{\pi r^{2}}$$,\frac{\rho \left(L_{3}+L_{2}/2\right)}{\pi r^{2}}$$,\frac{\rho L_{o}}{\pi r_{e}^{2}}$,	*R*_1_ = 5.8984 × 10^7^ *R*_2_ = 4.0065 × 10^7^ *R*_3_ = 3.4867 × 10^7^
Current source *i_s_*	$i_{s}=\frac{\partial V(t)}{\partial t}$	

## Data Availability

No new data were created or analyzed in this study. Data sharing is not applicable to this article.

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
