# Peer review of "Waveform Design Method for Piezoelectric Print-Head Based on Iterative Learning and Equivalent Circuit Model"

_micromachines, 2023, doi:10.3390/mi14040768_

Round 1
Reviewer 1 Report
This work claims that the flow rate of the fluid at the nozzle determines the formation process of droplets. Works was made to optimize the waveform aiming to suppress the residual vibration and minimize the droplets.
There are some questions for the authors.
1, in sedion 3 , the authors spend lots of words for the Equivalent circuit model of the inkjet nozzle. The author should give a clear description why they give a introduce of the Equivalent circuit model, and how this model was related to the Iterative learning method.
2, How does the Iterative learning method optimize the waveform? Currently, it is difficult for the reader to understand the working principles of this Iterative learning method.
3, in Figure 5, the vertical coordinates is the volume flow rate, however, drive waveform and Weight which has different unit with the flow rate is also drawn on this figure. It is unsuitable.
4, How the volume flow rate In Figure 5 and Figure 6a was obtained? By calculation? Or by experiential methods? If by calculation, how to verify the reliability of this flow rate? The flow rate at the orifice should be determined by the pressure just behind the orifice. However, the reader cannot fine calculated or experimental measured pressure behind the orifice, or the pressure wave propagation in the channel of the inkjet nozzle.
5, the authors claimed that the residual vibration will limit the printing frequency. And by using their optimized waveform, the residual vibration can be suppressed. However, the authors did not give data about the maximum printing frequency by using their optimized waveform.
6, in Figure 6b and Figure 9b, the shape of the optimized waveform is quite complex. The reader cannot find how this waveform was generated in this manuscript? As far as I know, it should be a challenge for the driver of MicroFab to generate such a waveform.
7, swallow the second droplets by a meniscus seems already reports by other researchers.
Author Response
Please see detailed responses in attachment.

Reviewer 2 Report
Review of “Waveform Design Method for Piezoelectric Print-head based on Iterative Learning and Lumped Parameter Model” for micromachines on 07/02/2023.
This manuscript proposes a waveform design method to control the volume flow rate at the nozzle, which is based on the iterative learning method. The research is conducted both theoretically and experimentally, which proves that the method is effective to design appropriate waveforms to suppress residual vibration with smaller droplets, the paper might be recommended for publication if following questions and comments are properly addressed.
(1) The section 2 on the experimental setup might be put into section 6, because they are both related to the experimental investigation.
(2) It is worthy to examine the purpose on using an equivalent model of the circuit to represent the PPH system; this system can be directly described by a multi-dimensional second-order differential equation array.
(3) The control algorithm is designed with an iterative manner, however, it would be better to demonstrate the efficiency of the optimization, especially if the control is in real-time.
(4) Some quantitative results can be presented in the conclusion section.
(5) Perhaps an abbreviation table can be added into the manuscript.
Round 2
Reviewer 2 Report
I am fine with the revision. Thanks